# Vitamin D and ω-3 Supplementations in Mediterranean Diet During the 1st Year of Overt Type 1 Diabetes: A Cohort Study

**DOI:** 10.3390/nu11092158

**Published:** 2019-09-09

**Authors:** Francesco Cadario, Erica Pozzi, Stefano Rizzollo, Marta Stracuzzi, Sheila Beux, Alberto Giorgis, Deborah Carrera, Fabiola Fullin, Sergio Riso, Angela Maria Rizzo, Gigliola Montorfano, Marco Bagnati, Umberto Dianzani, Philippe Caimmi, Gianni Bona, Camillo Ricordi

**Affiliations:** 1Division of Pediatrics, University of Piemonte Orientale, 28100 Novara, Italy; 2IRCAD (Interdisciplinary Research Center of Autoimmune Diseases), 28100 Novara, Italy; 3Diabetes Research Institute Federation, Miami, FL 33163, USA; 4Department Dietetic and Clinical NutritionUniversity Hospital of Novara, University of Piemonte Orientale, 28100 Novara, Italy; 5Faculty of Agricultural and Food Sciences, UniversitàdegliStudi di Milano, 20133 Milan, Italy; 6Department of Pharmacological and Biomolecular Sciences (DiSFEB), Laboratory of Membrane Biochemistry and Applied Nutrition, UniversitàdegliStudi di Milano, 20133 Milan, Italy; 7Clinical Biochemistry, University Hospital of Novara, 28100 Novara, Italy; 8Immunology, Department of Health Sciences, Università del Piemonte Orientale, 28100 Novara, Italy; 9Medical Direction University Hospital of Novara, 28100 Novara, Italy; 10Diabetes Research Institute and Cell Transplant Center, University of Miami Miller School of Medicine, Miami, FL 33136, USA

**Keywords:** cholecalciferol, omega3, EPA, DHA, arachidonic acid, AA/EPA ratio, type 1 diabetes, remission period, honeymoon period

## Abstract

Vitamin D and omega 3 fatty acid (ω-3) co-supplementation potentially improves type 1 diabetes (T1D) by attenuating autoimmunity and counteracting inflammation. This cohort study, preliminary to a randomized control trial (RCT), is aimed at evaluating, in a series of T1D children assuming Mediterranean diet and an intake of cholecalciferol of 1000U/day from T1D onset, if ω-3 co-supplementation preserves the residual endogen insulin secretion (REIS). Therefore, the cohort of 22 “new onsets” of 2017 received ω-3 (eicosapentenoic acid (EPA) plus docosahexaenoic acid (DHA), 60 mg/kg/day), and were compared retrospectively vs. the 37 “previous onsets” without ω-3 supplementation. Glicosilated hemoglobin (HbA1c%), the daily insulin demand (IU/Kg/day) and IDAA1c, a composite index (calculated as IU/Kg/day × 4 + HbA1c%), as surrogates of REIS, were evaluated at recruitment (T0) and 12 months later (T12). In the ω-3 supplemented group, dietary intakes were evaluated at T0 and T12. As an outcome, a decreased insulin demand (*p* < 0.01), particularly as pre-meal boluses (*p* < 0.01), and IDAA1c (*p* < 0.01), were found in the ω-3 supplemented group, while HbA1c% was not significantly different. Diet analysis in the ω-3 supplemented group, at T12 vs. T0, highlighted that the intake of arachidonic acid (AA) decreased (*p* < 0.01). At T0, the AA intake was inversely correlated with HbA1c% (*p* < 0.05; *r*;. 0.411). In conclusion, the results suggest that vitamin D plus ω-3 co-supplementation as well as AA reduction in the Mediterranean diet display benefits for T1D children at onset and deserve further investigation.

## 1. Introduction

Nutraceutic is a neologism made up of the words “nutrition” and “pharmaceutical” which was coined in 1989 by Stephen de Felice to label the science that focuses on nutritional principles contained in foods and their beneficial effects on health. This discipline has its roots in the not-so-recent past, but it has taken on scientific value over the last fifty years. However, it faces inherent difficulties in the assessment of clinical outcomes for a specific nutrient outside of the whole nutrition [1], which would be relevant for management of specific diseases. Specific nutrients, such as vitamin D and omega-3 polyunsaturated fatty acids (ω-3), are of concern in type 1 diabetes (T1D), a chronic autoimmune disease caused by progressive selective destruction of pancreatic β-cells producing insulin, starting in infancy with the appearance of specific autoantibodies, and with clinical outbreak during the pediatric age range in 90% of cases. Hyperglycemia is a characteristic sign of the disease and it requires a precise administration of insulin throughout life in order to restore metabolic control, but it does not abolish disabling complications over time. T1D entails a great deal of individual, familial, and social commitment. The rise in incidences of T1D in most developed countries, and its shift towards younger ages of life, implies that a genetic predisposition must be associated with environmental factors, including nutrients or other potentially reversible causative agents.

In 2003, Lars C. Stene found a low occurrence of T1D in the coastal areas of Norway with greater availability of fish, compared to areas in the hinterland [2]. This finding suggested a role of ω-3 and vitamin D as possible environmental determinants of the disease. The prevention capability of 2000 IU of vitamin D consumed daily over the first year of the patient’s life was reported by Elina Hyppönen et al. in 2001 [3]. Surprisingly, very few of such innovative studies have become of clinical utility in the prevention or treatment of T1D, except for a consensus on correcting vitamin D deficiency [4]. Recently in TEDDY, a large network cohort study, vitamin D or ω-3 supplementation during pregnancy did not confer any effect in the development of islet autoantibodies [5]. Conversely, in the first years of overt disease, vitamin D and ω-3 supplementation have been found to play a clinical role, as shown by SEARCH, two large prospective nutritional studies, reporting that the amount of ω-3 in the diet correlates inversely with glycosylated hemoglobin (HbA1c%) and positively with persistent fasting C-peptide (FCP) [6,7].

A possible therapeutic role of vitamin D in T1D might be mediated by its regulatory activity on the immune system and the autoimmune response through vitamin D receptors expressed on both immune cells and pancreatic β-cells. The first clinical evidence of this in T1D was given by Treiber et al., with the administration of high doses of cholecalciferol (70 IU/kg/day) in a randomized controlled trial on children with a recent onset of the disease, showing a capacity to restore immunosuppressive Treg cells and potentially counteract the elective autoimmune damage to pancreatic β-cells [8]. In this study, the supplementation with vitamin D did not display significant improvement on stimulated C-peptide (SCP). Recently, in a study of Iranian children at the onset of overt T1D, Habibian et al. provided evidence of vitamin D as an environmental factor decreasing the fall of C-peptide in children with recent T1D [9].

The ability of ω-3 to counteract the pathogenic pathways of T1D has been highlighted in two studies showing that its nutritional intake may reduce blood glucose levels [10] and limit β-cells apoptosis mediated by glucolipotoxicity [11]. Specifically, diet supplementation with ω-3 was found to lead to reduction of postprandial glycemia and improvement of glycemic variability, mediated by the inhibition of neoglucogenesis [10], and counteracting β-cell apoptosis through activation of the *Eovl2*/docosahexaenoic acid(DHA) enzyme axis [11]. These reports suggest both an immunologic and metabolic role for ω-3, limiting the post-prandial blood glucose increase, and protecting β-cell apoptosis induced by glucolipotoxicity. The role of ω-3 and omega-6 polyunsaturated fatty acids (ω-6) in the regulation of autoimmunity in NOD mice is detailed in a study byXinum Bi et al. [12]. In this work, eicosapentenoic acid (EPA) and DHA reduced the imbalance of Th1/Th2 cells and the proportion of Th17 cells, and increased Treg. In contrast, the ω-6 arachidonic acid (AA) intake increased the number of Th17, without significant difference in the count of Treg cells. In the same direction, ω-3 supplementation decreased levels of IFN-Υ, IL-17, IL-6, and TNF-α, supporting its anti-inflammatory ability.

The ideal point in time for testing a possible role of vitamin D and ω-3 should be at clinical onset of T1D when, after the start of insulin therapy and the achievement of stable metabolic compensation, about 80% of children and adolescents experience a partial remission, reducing insulin demands to maintain euglycemia. Afterwards, a resumption of the autoimmune process determines a progressive growth of insulin requirements and brings the so-called “honeymoon phase” of T1D [13] to an end. Sustaining the recovery of endogen insulin secretion (REIS) should be the ultimate goal of T1D therapy in order to reduce severe hypoglycemia and avoid its long-term complications [14].

Given that insulin therapy is the first factor conditioning the restoration of the critical mass of β-cells during the honeymoon phase through stabilization of blood glucose, to date there is limited experience in the effect of combined supplementations with vitamin D and ω-3 within the first year of overt disease in children. Since anecdotal cases have suggested that this co-supplementation may prolong REIS [15,16,17], we looked at the introduction, in a series of T1D children already assuming Mediterranean diet and supplementation of 1000 IU of cholecalciferol from onset, of an additional ω-3 supplement starting from the first semester of overt disease and evaluate outcomes as a pilot study before further randomized controlled intervention trials (RCT).

## 2. Materials and Methods

Intervention group: Starting in January 2017 we prospectively studied 26 new consecutive T1D onsets referred to the pediatric diabetes service of the University Hospital of Novara (AOU-Novara). Everyone received training on the Mediterranean diet, a cholecalciferol supplementation of 1000 IU/day, and within the first semester of overt disease, also was entered to ω-3 supplement of ultra-refined fish oil, equivalent to 60 mg/kg/day of eicosapentenoic acid (EPA) and docosahexaenoic acid (DHA).

Retrospective controls: All patients with T1D onset in 2014–2016 were admitted to the study as controls, after having received their parents’ written consent. One patient (n = 38) was excluded since he was the anecdotic case who received ω-3 supplementation before this study. Data were structured in a database. The patients had been introduced to the Mediterranean diet (as reported in the Appendix A) and 1000 UI/day cholecalciferol supplementation since the first month of clinical disease, without a ω-3 supplement.

Inclusion and exclusion criteria: Patients with renal cysts or affected by sarcoidosis, histoplasmosis, hyperparathyroidism, lymphoma, and tuberculosis were excluded. T1D patients with thyroiditis and celiac disease following a gluten-free diet and who were tissue transglutaminase autoantibody (tTGA) negative were not excluded. Patients treated with drugs that could affect immunity or glucose metabolism, including corticosteroids, ciclosporin, and tacrolimus, were excluded.

Recruitment for the study took place from 1 January 2017 to 31 December 2018. The intervention lasted one year for each case. This study was approved by the ethics committee of AOU Novara, and all patients’ parents signed the informed consent form. This trial was registered at the ClinicalTrials.gov website (identifier: NCT03911843).

Diagnosis of T1D was performed according to the American Diabetes Association criteria [18]. Micro or macro vascular complications were defined according to the ISPAD criteria [19]. Children were evaluated using Italian growth charts [20]. At the onset of the disease, data on the presence of diabetic ketoacidosis, HbA1c%, insulin requirements, 25(OH)D levels, thyroid function, antibody titles of GADA, IAA, and IA-2, antibody patterns of celiac disease, and lipid profiles were collected. Evidence of ketoacidosis was assessed according to the ISPAD criteria [19], and severe DKA was considered if pH ≤7.1. The vitamin D status, as 25(OH)D levels, sufficiency, deficiency or insufficiency, were defined according to the Endocrine Society criteria, graded as sufficiency >30 ng/mL(>75 nmol/l), insufficiency 21–29 ng/mL(52.5–72.5 nmol/l), and deficiency ≤20 ng/mL(≤50 nmol/l) [21,22]. The annual screening (blood count, lipid profile, urine test, thyroid function, and aPTT coagulation tests) was performed and evaluated at recruitment(T0) and twelve months later(T12). Blood glucose meters (Conturnext USB^®^ Ascensia Diabetes Care) were supplied to the patients enrolled to standardize blood glucose measurements, and their data were downloaded with dedicated software (Glucofacts^®^ Ascensia Diabetes Care, FreeStyleLibreLink^®^, Medtronic CareLink Pro^®^, Accu-ChekChekSmartPix^®^, Diasend^®^ or Dexcom Clarity^®^). Clinical data, insulin demand, and laboratory analysis results were entered in a structured database.

### 2.1. Supplementations

Supplementation of ω-3 was considered the intervention (T0). The preparation administered was a highly purified fish oil used in order to avoid pollutants, containing a mixture of omega 3 long chain fatty acids (ω-3 LCFA) standardized for contents of eicosapentaenoic acid (EPA) and docosahexaenoic acid (DHA), in capsules or in liquid form (Ener Zone Omega 3 RX^®^ Equipe Enervit). A liquid preparation was used in the case of difficulties in swallowing capsules or concomitant celiac disease because it is certified as gluten-free. The preparations contained antioxidants to preserve ω-3 LCFA, tocopherol (1 mg in 1 g of ω-3 LCFA), palmitate, and rosemary extract. EPA and DHA at 60 mg/kg/day were administered for 12 months. The supplementation goal was an AA/EPA ratio between 1.5–3. The investigation of AA/EPA ratios was performed in cases at enrollment (T0), and repeated after 3 (T3), 6 (T6), and 12 months (T12).

Cholecalciferol supplementation was fixed at 1000 IU/day (25 mcg/day), both in patients receiving supplementation and controls [21]. Vitamin D was determined as 25(OH)D level in patients receiving supplementation and controls at the clinical onset of T1D and at enrolment (T0). For patients receiving supplementation this continued at T3 and T12.

### 2.2. Diet

At T1D clinical onset, a dietician provided a training courseon the Mediterranean diet [23]. The educational items were standardized [24] (see Appendix A). A dietitian’s counseling was provided to each patient and accompanying parent every 3 months in order to sustain a Mediterranean diet over time. At T0, written materials were provided on the foods’ main source of PUFA, to sustain the intake of ω-3-rich foods (blue fish, nuts, walnuts, poultry, eggs, olive oil, flax seeds, and leafy vegetables) and limit food rich in ω-6 (beef, pork, and offal).

At T0 and T12, dietary intakes were assessed based on a weekly food diary and further interviews to determine macro and micronutrients intake as precisely as possible. The amounts of the nutrients have been evaluated with the support of the Food Composition Database for Epidemiological Studies (BDA version 1-2015) [23], assessing protein (g/day), energy (Kcal/day), cholesterol (mg/day), carbohydrates (g/day), dietary fiber (g/day), vitamin D (g/day), PUFA (g/day), arachidonic acid (AA g/day), EPA (g/day), and DHA (g/day).

### 2.3. Assays

Plasma glucose levels (mg/dl; 1 mg/dl:0.05551 mmol/liter) were measured by the gluco-oxidase colorimetric method (GLUCOFIX^®^, by Menarini Diagnostics, Florence, Italy). HbA1c% levels were measured through high-performance liquid chromatography (HPLC), using a Variant machine (Biorad, Hercules, CA) Intra- and inter-assay coefficients of variation were, respectively, lower than 0.6% and 1.6%. Linearity was excellent, from 3.2% (11 mmol/mol) to 18.3% (177 mmol/mol). The presence and titration of antibodies GAD65, IA2 and IAA, expressed in IU/mL, was carried out by immunoradiometric assay (IRMA), with analytical coefficients of analysis of 13%, 8.4%, and 13%, respectively. The semi-quantitative determination of TGA, AGAD-G, and AGAD-A, expressed in IU/mL, was carried out on serum by QUANTA Flash, a rapid response immunochemiluminescence test (CIA) performed on the BIO-FLASH instrument. The coefficients of analytical variability were, respectively, 5.5%, 6.7%, and 4.3%. The quantitative determination of the anti-TG and anti-TPO autoantibodies, expressed in IU/mL, of the TSH, expressed in µUI/mL, and of the fT4, expressed in ng/dl, was performed by a competitive immunoassay using a direct immunochemiluminescence technique, using the Siemens ADVIA centaur XPT system.

The circulating vitamin D levels were measured by serum 25(OH)D assay, using an immunochemiluminescence (CLIA) method (DiaSorin Liaison Test^®^, Stillwater, MN, USA). The coefficient of inter-individual variability of the method (CVa%) was 10%.

Fatty acids (AA, EPA, DHA percentages, and respective AA/EPA ratio) were determined by high-resolution capillary gas chromatography in whole blood, using dried blood spots testing [25].

The levels of circulating C-peptide, expressed in ng/mL, were measured, on citrate or heparinized plasma, both by chemiluminescent “sandwich” immunoassay (DiaSorin Liaison^®^) and by immunochemiluminescence with the automatic analyzer Immulite 2000 medical system, with a coefficient of variability of 7.4%.

To asses REIS, we used a composite index of insulin demand adjusted for metabolic control IDAA1c (calculated as insulin dose (UI/kg/24 h) × 4 + HbA1c%). A score of <9 met the definition of partial remission and REIS, according to the previous studies in TrialNet [26].

### 2.4. Statistical Analysis

Data were expressed as mean ± SD of absolute values. The differences between groups were evaluated for the continuous variables through Mann-Whitney *U* tests. Chi-square tests were used to compare the nominal variables between groups. In supplemented subjects, the evaluation of variation between T0 and T12 for all metabolic parameters was performed with *t-*test for repeated measures. The association between the variables was evaluated according to a Pearson test, after proper logarithmic transformation of the parameters, if required. Trend evaluation across 25(OH)D levels was performed at the onset though multinomial regression analyses. Moreover, in supplemented cases, a trend evaluation of metabolic parameters across each time point and timing of supplementation (T0 → T12) was performed though multinomial regression analyses. Significant *p*-values were less than 0.05. All statistical analyses were performed using SPSS 22.0 (SPSS Inc., Chicago, IL, USA).

### 2.5. Patients

At T0, we analyzed 64 patients (26 supplemented cases and 38 controls, Table 1), then the analysis during the time of the trial was carried out on 59 patients, whose observations (HbA1c%, average daily insulin needs (IU/Kg/day) and IDAA1c) were available from recruitment (T0), to 3, 6 and 12 (T12) months. Two cases were dropped because they were not adhering to therapy, two cases stopped because of ω-3 side effects, and one control was dropped because he changed residence. Finally, 59 patients were evaluated, 22 as supplemented cases and 37 as controls. Only the supplemented cases received ω-3 supplementation, as mentioned above.

## 3. Results

At onset of T1D:Between the whole series (64) of T1D children, we found in 82.5% of cases a vitamin D insufficiency [≤30 ng/mL (≤75 nmol/L)] in T1D children at time of clinical onset of T1D. A severe deficiency ≤ 10 ng/mL (25 nmol/L) was present in 12.7%, and those ones displayed a significantly lower FCP level (*p* < 0.02) and pH (*p* < 0.02) than the others. Variability in pH (*p* < 0.01) and FCP (*p* < 0.02) across vitamin D levels was observed.

Patients supplemented vs. not supplemented: At 12 months of ω-3 supplementation (T12), the cases (*n* = 22) showed significantly lower insulin needs than the controls (*n* = 37). In particular, lower daily insulin needs (0.49 ± 0.24 vs. 0.63 ± 0.19 IU/Kg/day; *p* < 0.01) and pre-meal bolus (0.22 ± 0.16 vs. 0.34 ± 0.14 IU/Kg/day: *p* < 0.01) were found, without differences in HbA1c% (NS). Analysis of the IDAA1c index at T12 showed IDAA1c < 9, consistent with a partial remission, in 12 of 22 (54.5%) cases vs. 7 of 37 controls (18%; *p* < 0.01) (Figure 1, Table 2 and Table 3).

### The Impact of the Diet

The evaluation of diet composition was made only in cases receiving the ω-3 supplementation, comparing diet intakes at T0 and T12. It showed that within macronutrients, carbohydrate (*p* < 0.09), fiber (*p* < 0.05), and protein (*p* < 0.05) intakes were lower at T12 then T0. The micronutrients in the diet were similar for vitamin D, EPA, and DHA (NS). However, the diet intake of AA was significantly lower (0.25 ± 0.1 vs. 0.20 ± 0.1 g/die; *p* < 0.01) at T12 vs. T0. The caloric intake was similar at the beginning and at the end of the one-year ω-3 administration (NS) (Table 4).

Correlations: At disease onset, 25(OH)D levels were correlated with pH (*r*: 0.359, *p* < 0.01). HbA1c% values were correlated with insulin daily requirement (*r*: 0.568, *p* < 0.0001) and weakly inversely correlated with pH (*r*: −0.245, *p* = 0.05). The AA intake at starting point of ω-3 supplementation (T0) was inversely correlated with HbA1c% (*p* < 0.05, *r*: −0.411). After 12 months of ω-3 supplementation, vitamin D was inversely correlated with HbA1c% (*p* < 0.05, *r*: −0.462).

Limitations: There are several limitations to conclude on the efficacy of ω-3 in T1D children. (1) The study was not randomized; it is therefore a preliminary study to be finalized by subsequent RCTs. (2) The IDAA1c index is a surrogate of REIS and not a direct evaluation of insulin secretion. (3) Controls are retrospective, so the comparability of series concerns only some data, such as HbA1c%, insulin needs, and IDAA1c, but not FCP, AA/EPA, and nutritional intakes. (4) The target AA/EPA levels weren’t reached with the doses assigned (Table 2).

Side effects: One female child reported diarrhea, so she stopped the fish oil supplement with a quick return to normality. A female teenager with preexisting thyroiditis presented a transient suppression of TSH, which returned to normal values three months after ω-3 suspension. One male child at T12 showed a lengthening of clotting time (aPTT), the parameters returned gradually to the norm after suspension of ω-3 supplement without clinical signs of hemorrhage. No other side effects have been reported, no hemorrhagic clinical symptoms or manifestations that could lead one to suspect coagulation involvement.

## 4. Discussion

In the management of T1D, there is a general agreement that the Mediterranean diet improves metabolic control [27], and the administration of vitamin D supplements avoids its deficiency [4]. A recent systematic review of vitamin D supplementation at the onset of T1D concluded that alphacalcidole and cholecalciferol supplementations have beneficial effects on daily insulin doses, HbA1c%, FCP, and SCP. In addition, it indicates that further randomized controlled trials based on biomarkers are needed to define their optimal contributions [28]. However, there are no directions to date on supplementation of ω-3 in young T1D subjects, despite some recent anecdotal data showing persistent partial remissions resulting from vitamin D and ω-3 co-supplementation introduced early after the disease onset [15,16,17]. Beyond these anecdotal cases, there is little scientific work on clinical outcomes of co-supplementation of vitamin D and ω-3 in recent-onset T1D. In this study involving a series of T1D children, already assuming a Mediterranean diet and cholecalciferol, the additional supplement of ω-3 in recent-onset cases seemed to have beneficial effects, which suggests the need to design further RCTs.

Our previous findings in this context showed a widespread insufficiency of vitamin D at the clinical onset of T1D and a relationship between its severe deficiency with reduced FCP and pH at clinical onset, in line with the reports of other authors [29,30,31]. This highlights the importance of vitamin D status in childhood to determine the severity of T1D at clinical onset. Ultimately, it is mandatory to correct vitamin D deficiency from the onset of T1D.

In our study, the daily supplementation of ω-3 starting from clinical onset led to, one year later, less insulin demand without affecting glycemic control, since the subjects had similar HbA1c%. Considering IDAA1c (insulin dose adjusted for HbA1c%) as a surrogate index of residual β-cell function, and IDAA1c ≤ 9 is indicative of partial remission of T1D [25], our data show that ω-3 may preserve β-cell secretion. This finding, if confirmed by further RCTs, might be considered for assisting children beginning at the onset of the disease. The clinical outcome of a reduced insulin demand, mainly at meals, is compatible with the hypothesis of the inhibition of postprandial protein neoglucogenesis [10]. Interestingly, the effects of co-supplementation of vitamin D and ω-3 have also been reported in randomized clinical trials outside of childhood T1D, on gestational diabetes and on multiple sclerosis [32,33]. Those studies reported a reduction of blood glucose and fasting plasma insulin, and an increase of insulin sensitivity. Particularly, the trial on gestational diabetes highlighted a synergism of vitamin D and ω-3 [33]. While clinical randomized trials on co-supplementation of vitamin D and ω-3 in T1D are ongoing, our data suggest a benefit, and aim at finalizing further randomized investigations to the topic.

Despite the apparent benefits of reduced insulin demand in those who took ω-3 supplements, in our study, we found similar HbA1c% at comparable points from the onset of disease for both supplemented cases and controls, which represents a similar outcome to metabolic balance. This likely occurred with the addition of insulin doses in controls to keep their blood sugar within limits. Unexpectedly, we found a decreasing trend of fasting C-peptide (FCP) from the start of ω-3 supplement (T0) to one year later (T12). A plausible explanation for similar metabolic control with less insulin administration, despite decreased FCP, is probably related to the counteraction of ω-3 on neoglucogenesis, which limits postprandial glucose increase and reduces insulin need for meals. The lowering of the blood glucose excursions could in turn affect the process of apoptosis of β-cells reducing glucolipotoxicity, and so could preserve REIS, according to findings in translational models [11]. This assumption might be investigated through analysis of stimulated C-peptide (SCP) after a standardized mixed-meal tolerance test, which allows for a direct measurement of insulin secretion, in future RCTs. A possible immune regulatory role of ω-3 supplementation could concur, together with vitamin D, to improve T1D, and should be investigated by assessing biomarkers of autoimmunity (e.g., lymphocyte subpopulations Treg or Th-17). This study could not determine if the decreased insulin demand is sustained only by reduced needs or also by preserved REIS over time, because IDAA1c is a composite index of insulin needs and metabolic control, but not a direct measurement of REIS.

Another limit of this study is the failure to achieve the target levels of the AA/EPA ratio, probably due to reduced compliance of pediatric patients in the assumption of fish oil, or insufficient dosage. Reaching an AA/EPA range 1.5–3 will therefore be a goal in future trials.

The diet analysis in the ω-3 supplemented group, showed a decrease in carbohydrates, from T0 to T12, which although not significant, may have played part in the reduction of pre-prandial insulin demands. Protein and fiber moderately decreased, and no significant changes were found in terms of total lipids. EPA and DHA in the diet were unmodified and not significantly increased, respectively. The reduction of AA intake was significant and relevant. It is likely that the AA reduction takes part in determining decrease of AA/EPA and outcomes. This might be related to a competition between ω-3 and ω-6 for the common metabolic pathway of LCFA and suggests that a reduction of AA nutritional intake could improve the results of omega 3 supplementation and it should undergo future research in further studies.

## 5. Conclusions

Given that this study is a preliminary to future RCTs, our findings suggest that a ω-3 supplement, in the context of a Mediterranean diet and vitamin D administration, gives benefits to children with T1D, with a reduction of insulin requirements after 12 months of supplementation, despite a decrease in FCP. Vitamin D and dietary intake of ω-3 and AA play a role in whether metabolic control becomes better or worse, and perhaps whether favorable or unfavorable autoimmunity follows. Particularly, further investigations should consider the direct biomarkers of insulin secretion (such as SCP) and of T-cell immune-regulatory subgroups (such as Tregs, Th17), to define if AA and ω-3 assign metabolic or immune effects, or both.

## Figures and Tables

**Figure 1 nutrients-11-02158-f001:**
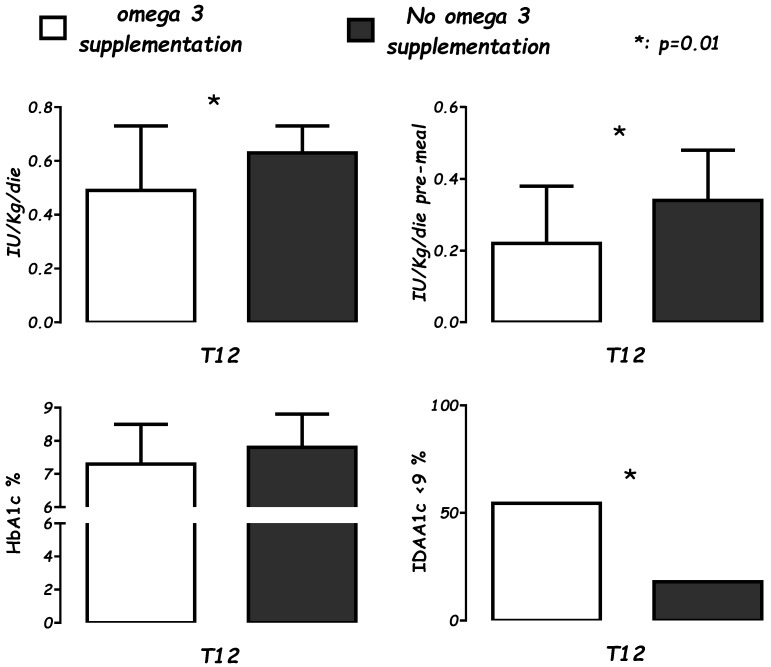
Levels of average daily insulin needs (IU/Kg/day), pre-meal boluses (IU/Kg/day), glicosilated hemoglobin percentage (HbA1c%), and the Insulin Dose adjusted for glicosilated hemoglobin percentage partial remission index (IDAA1c ≤ 9), in supplemented case (white), for 22 patients after 12 months of ω-3 supplementation vs. controls (black), a group of 37 not-supplemented patients at the same time.

**Table 1 nutrients-11-02158-t001:** Characteristics of participants at baseline. Data presented as mean + SD or percentages as appropriate.

	Cases	Controls	*p* Value
*n*	26	38	
Gender (female/male)	14/12	20/17	
Age (years)	8.7 ± 4.6	8.8 ± 3.6	NS
Body weight (Kg)	30.7 ± 17.5	32 ± 14	NS
BMI (Kg/m^2^)	−0.92 ± 1.1	−1.1 ± 2	NS
HbA1c%	11.3 ± 2.2	11.6 ± 2.6	NS
Insulin dose (UI/Kg/day)	0.61 ± 0.22	0.69 ± 0.28	NS
CSII/MDI device	2/24	11/27	

BMI (Body Mass Index), HbA1c% (glycosylated Hemoglobin in percentage), CSII (Continuous Subcutaneous Insulin Injection), MDI (Multiple Daily Injections).

**Table 2 nutrients-11-02158-t002:** Metabolic parameters in omega-3 supplemented cases, at enrolment T0, and after three (T3), six (T6), twelve (T12) months of supplementation. Data are expressed as mean ± SD. The significance among the four measures (T0, T3, T6 and T12) or three measures (T0, T3, T12) was calculated by Friedman test.

	T0	T3	T6	T12	*p^for trend^*
C-peptide (ng/mL)	0.9 ± 0.7 ^a^	0.5 ± 0.3	-	0.38 ± 0.5 ^a^	*p* < 0.01
Mean Glucose (mg/dl)	140 ± 26	155 ± 48	148 ± 38	141 ± 34	NS
SD	50 ± 18	45 ± 24	50 ± 24	57 ± 25	NS
Vitamin D (nmol/L)	31 ± 12 ^b^	41 ± 14	-	35 ± 12 ^b^	*p* < 0.001
AA/EPA	55 ± 38 ^a^	6.8 ± 5	-	8.8 ± 6 ^a^	*p* < 0.0001

a = *p* < 0.001 T0 vs. T12; b = *p* < 0.01 T0 vs. T12. HbA1c% (glycosylated hemoglobin percentage), AA/EPA (Arachidonic Acid and Eicosapentenoic Acid ratio).

**Table 3 nutrients-11-02158-t003:** Metabolic parameters and insulin requirement in Supplemented with omega-3 fatty acids (3) vs. Not Supplemented patients, after 3 (T3), 6 (T6) and 12 (T12) months of 3 supplementation.

Timing					*p* Value
T0	*n*	S	*n*	NS	
Insulin (IU/Kg/day)	26	0.61 ± 0.22	38	0.69 ± 0.28	NS
HbA1c (%)		11.3 ± 2.2		11.6 ± 2.6	NS
T3	*n*	S	*n*	NS	
Insulin (IU/Kg/day)	22	0.37 ± 0.3	37	0.41 ± 0.2	NS
Insulin bolus (IU/Kg/day)		0.17 ± 0.1		0.22 ± 0.1	NS
Insulin basal (IU/Kg/day)		0.20 ± 0.1		0.19 ± 0.1	NS
HbA1c (%)		7.3 ± 1.1		7.6 ± 0.1	NS
IDAA1c < 9 (%)		50%		53%	NS
T6	*n*	S	*n*	NS	
Insulin (IU/Kg/day)	22	0.44 ± 0.2	37	0.51 ± 0.2	NS
Insulin bolus (IU/Kg/day)		0.20 ± 0.1		0.27 ± 0.1	0.06
Insulin basal (IU/Kg/day)		0.24 ± 0.1		0.23 ± 0.1	NS
HbA1c (%)		7.4 ± 1.1		7.5 ± 1.1	NS
IDAA1c < 9		50%		34.1%	NS
T12	*n*	S	*n*	NS	
Insulin (IU/Kg/day)	22	0.49 ± 0.2	37	0.63 ± 0.1	<0.01
Insulin bolus (IU/Kg/day)		0.22 ± 0.1		0.34 ± 0.1	<0.01
Insulin basal (IU/Kg/day)		0.29 ± 0.1		0.28 ± 0.09	NS
HbA1c (%)		7.4 ± 1		7.8 ± 1.0	NS
IDAA1c < 9		54.5%		18.9%	<0.01

Data are expressed as mean ± SD. HbA1c % (glycosylated hemoglobin percentage). IDAA1c < 9 (insulin dose adjusted for glycosylated hemoglobin‘percentage index, <9 was considered a partial remission).

**Table 4 nutrients-11-02158-t004:** Diet composition of supplemented cases at enrollment and after 12 months of ω-3 supplementation. Eicosapentanoic acid (EPA), docosahexaenoic acid (DHA), arachidonic acid (AA), polyunsaturated fatty acids (PUFAs), carbohydrates (CHO), fibers, and proteins are expressed in g/day. Vitamin D intake is expressed in μg/day. Data are presented as mean, standard deviation (DS), and statistical significance (*p*).

Diet Intake	T0	T12	*p*
Caloric intake (Kcal/day)	1739 ± 556	1576 ± 463	0.09
Protein g/day	73 ± 21	61.4 ± 20	<0.05
Lipid g/day	66 ± 16	64 ± 18	NS
CHO g/day	221 ± 87	190 ± 73	0.09
Fiber g/day	18 ± 7.3	14 ± 4.6	<0.05
Vitamin D µg/day	4 ± 2.7	4.6 ± 1.9	NS
PUFAs g/day	8.6 ± 3.9	10.8 ± 6.5	NS
AA g/day	0.25 ± 0.1	0.20 ± 0.1	<0.01
EPA g/day	0.22 ± 0.1	0.22 ± 0.1	NS
DHA g/day	0.35 ± 25	0.43 ± 0.22	NS

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
