# Peer review of "Vitamin D and ω-3 Supplementations in Mediterranean Diet During the 1st Year of Overt Type 1 Diabetes: A Cohort Study"

_nutrients, 2019, doi:10.3390/nu11092158_

Round 1

Reviewer 1 Report

Dear authors,

Your study exploring whether omega3 supplementation improves the clinical outcomes of type I diabetes is pretty compelling.  However, I have some concerns, which if addressed would further clarify the manuscript. 

Were the cases and controls given both vitamin D and omega3? It seems from Table2, they were given both..but the conclusions of the results do not mention that.

I would suggest that for table2, do a prospective comparison with the controls? How does the Insulin demand in controls compares with the Insulin demand in cases?What does the P-value indicate here? I would suggest do a comparison of each time-point with controls.

Authors should comment on no change in the HB1Ac values.

According to table3, the carbohydrate composition of the diet was significantly lower in the cases. Could that be a reason for reduced Insulin requirement?

Author Response

we are submitting the revised version of our review entitled “Vitamin D and type 1 diabetes” for possible publication in your journal. We particularly thank the Reviewers for the careful correction suggested. We are really grateful for the opportunity of improving our work and we hope to have satisfactorily answered to all questions.

Reviewer 1

Were the cases and controls given both vitamin D and omega3? It seems from Table2, they were given both..but the conclusions of the results do not mention that

Reply: Thank you for your comment. Both cases and controls were supplemented with cholecalciferol at the same dose (1000 IU/ day) from the onset of T1D. Only cases received omega 3 and cholecalciferol, while CONTROLS assumed cholecalciferol, without omega 3. We have modified Table 2 (pag 7) and added Table 3 (pag 7), where we showed the comparison between case and control subjects at different times, as suggested. Moreover, we have proceeded to a better clarify in the text, page 6, line 235-239: “After 12 months of co-supplementation of ω-3 and cholecalciferol, cases at T12 (n.22) showed a decrease of AA/EPA (55±38 vs. 8.8±6 ng/dl; p<0.01) and an increase of vitamin D (77.3.5±29 vs. 87.3.5±29 nmol/L; p<0.001) compared to cases at T0. Thus, the expected level of AA/EPA (1.5-3) was not achieved. The insulin demand increased (p<0.0001) and FCP decreased (p<0.0001) (Table 2)".

2) I would suggest that for table2, do a prospective comparison with the controls? How does the Insulin demand in controls compares with the Insulin demand in cases? What does the P-value indicate here? I would suggest do a comparison of each time-point with controls

Reply: Thank you for your suggestion. We added a comparison of each time-point with controls for metabolic parameters (in particular, insulin demand) in new Table 3 (pag 7).

3) Authors should comment on change in the HB1Ac values

Reply: Thank you for your suggestion. We add a sentence in the discussion section: “Despite the apparent benefit of the reduced insulin demand in whose assumed ω-3 supplementation,  we found in cases and controls similar HbA1c% at comparable points from onset of disease, which represents a similar outcome as metabolic balance. Likely, in controls it was related to additions of insulin doses to keep their blood sugar within limits."  at line 346-349, page 9.

4) According to table3, the carbohydrate composition of the diet was significantly lower in the cases. Could that be a reason for reduced Insulin requirement? 

Reply: Thank you for your comment. We have not comparable data for control subjects about the diet, given that controls were retrospectively recruited. Table 4 represents the food analysis only in cases at enrollment and after 12 months of omega 3 supplementation. The lower intake of carbohydrate at the end of the year of supplementation with omega 3 reported in the table, (p 0.09) is not statistically significant. Was considered significant p less than 0.05. We have better clarified in result section:“The evaluation of diet composition  was done only in cases receiving the ω-3supplementation, comparing diet intakes at T0 and T12. It showed within macronutrients, carbohydrate (p<0.09), fiber (p<0.05), and protein (p<0.05) intakes lower at T12 then T0. The micronutriens in the diet were similar for vitamin D, EPA, and DHA (p NS).  Instead, the diet intake of AA was significantly lower (0.25±0.1 vs. 0.20±0.1 g/die; p<0.01) at T12 vs. T0 at line 285-289 page 7.

Moreover we have changed in discussion: “From the diet analysis, in the ω-3 supplemented group, no significant changes were found from T0 to T12 in terms of nutrients (carbohydrates, lipids). The decrease of AA was significant and relevant.  Probably, the AA reduction concurred with ω-3 supplementation in determining decrease of AA/EPA and outcomes. This might be related to a competition between ω-3 and ω-6 for common metabolic PUFA pathways and it suggests that a better outcome might result if ω-3 supplementation  will be contextual to a reduction of AA in the diet“, at line 365-370 page 9.

Ultimately we believe the reduction of  carbohydrates found might concurred to lowering the blood glucose without determine the results (AA/EPA ratio, average daily insulin need, HbA1c%, and  IDDA1c), because to patients were not given indications to reduce carbohydrates in the diet, and we did not found relation with its amounts of  intakes both at T0 and T12.

Reviewer 2 Report

The manuscript “Vitamin D and ω-3 Supplementations in Mediterranean Diet during the 1st year of overt Type 1 Diabetes: a Cohort Study” submitted by Dr. Cadario and colleagues, the authors evaluated the effect of vitamin D and ω-3 co-supplementation from the new onset of T1D. The conclusion is that the co-supplementation in Mediterranean diet reduces the intake of arachidonic acid and may benefits to children with T1D. The authors also discussed the limitation of this preliminary study and future trial is expected. The study is scientifically sounds and well written. There are only a few comments aimed at improving the quality of the manuscript:

1. Results section, Line 231-232: The definition of vitamin D status is inconsistent with Line 139-141. Please clarify this point.

2. Line 236-237: The data and units of AA/EPA ratio and vitamin D are different with Table 2.

3. Table 2 and 3: Please change “die” to “day”.

4. Table 2 and 3: Why are there only ω-3 supplemented cases data and no data of controls?

5. Line 263: Please change “25OHD” to “25(OH)D”.

6. The authors determined 25(OH)D levels as vitamin D, but the description in the manuscript is confused with both words. The reviewer recommend to unify one to avoid confusion.

7. Line 329-331: The authors said there were no changes in caloric intake and nutrients after ω-3 co-supplementation, which is inconsistent with the results in Table 3. Please explain these data more carefully.

Author Response

Novara, 8th August 2019

Nutrients

Dear Editor,

we are submitting the revised version of our review entitled “Vitamin D and type 1 diabetes” for possible publication in your journal. We particularly thank the Reviewers for the careful correction suggested. We are really grateful for the opportunity of improving our work and we hope to have satisfactorily answered to all questions.

We also found in the first submitted version an incorrect data in table 1 (number of controls 38, not 37) that we corrected and highlighted. We apologize for this.

Thank you for your consideration,

Francesco Cadario,

Division of Pediatrics, Department of Health Sciences,

University of Piemonte Orientale,

Corso Mazzini 18, 28100, Novara, Italy.

Tel: +39 321 373 3279

Fax: +39 321 373 3598
